# Benchmark and Neural Architecture for Conversational Entity Retrieval from a Knowledge Graph

## ABSTRACT

This paper introduces a novel information retrieval (IR) task of Conversational Entity Retrieval from a Knowledge Graph (CER-KG). CER-KG extends non-conversational entity retrieval from a knowledge graph (KG) to the conversational scenario. The user queries in CER-KG dialog turns may rely on the results of the preceding turns, which are KG entities. Similar to the conversational document IR, CER-KG can be viewed as a sequence of interrelated ranking tasks. To enable future research on CER-KG, we created QBLink-KG, a publicly available benchmark that was adapted from QBLink, a benchmark for text-based conversational reading comprehension of Wikipedia. In our initial approach to CER-KG, we experimented with Transformer- and LSTM-based dialog context encoders in combination with the Neural Architecture for Conversational Entity Retrieval (NACER), our proposed feature-based neural architecture for entity ranking in CER-KG. NACER computes the ranking score of a candidate KG entity by taking into account a large number of lexical and semantic matching signals between various KG components in its neighborhood, such as entities, categories, and literals, as well as entities in the results of the preceding turns in dialog history. The experimental results for our initial approach to CER-KG reveal the key challenges of the proposed task along with the possible future directions for developing new approaches to it.

## CCS CONCEPTS

• **Information systems → Retrieval models and ranking**.

## KEYWORDS

Sequential IR, Entity Retrieval, Knowledge Graphs, Deep Learning, IR Benchmarks

**ACM Reference Format:**
Anonymous Author(s). 2023. Benchmark and Neural Architecture for Conversational Entity Retrieval from a Knowledge Graph. In *The Web Conference*. ACM, New York, NY, USA, 10 pages. https://doi.org/XXXXXXX.XXXXXXX

## 1 INTRODUCTION

The recent advances in deep learning have catapulted human-machine dialog from the narrow confines of scripted task completion into everyone's daily life. With the growing popularity of mobile devices and digital personal assistants, the human-machine dialog is also

well-poised to soon become the primary modality for information seeking. In conversational information seeking [11], users engage in a dialog with a search system to address their information needs. The user utterances in such dialog can take several forms, including queries and questions. Generating a search system's response for each form of user utterance in information seeking dialogues requires leveraging a wide variety of sources (text collections, knowledge graphs, tables, and databases) and an even wider variety of approaches that can utilize these sources along with the dialog context in the form of the preceding dialog turns.

Two major themes can be identified in prior research on conversational information seeking: conversational question answering (QA) and conversational information retrieval (IR). Conversational QA has been well-studied in the scenarios that utilize a textual collection [26, 42–45, 52], knowledge graph (KG) [8, 18, 24, 25, 35, 46, 48], table [23] and their combinations, such as KG and text [49, 50] or KG, text and tables [9]. Conversational IR research, however, has so far only focused on text collections [19, 31, 53], whereas **entity retrieval from a KG has not yet been studied in a conversational setting.** To address this oversight, we introduce a novel task of **C**onversational **E**ntity **R**etrieval from a **K**nowledge **G**raph (CER-KG) summarized in Figure 1 and defined as follows:

DEFINITION 1. *Conversational Entity Retrieval from a Knowledge Graph is an IR task that focuses on retrieving an entity or a set of entities in response to a free-form query that may explicitly or implicitly rely on the dialog context.*

This definition leads to several important differences between CER-KG and Conversational QA over a KG (CQA-KG). From a conceptual perspective, CER-KG extends entity retrieval from a KG to a dialog setting. Similar to conversational document IR [12], CER-KG can thus be viewed as a sequence of interrelated rounds of candidate KG entity retrieval and ranking. Correspondingly, the key challenges of CER-KG are the identification of comprehensive candidate entities in a KG and the discovery of effective relevance signals and methods to translate those signals into the accurate ranking of candidate entities. On the other hand, CQA-KG and QA from a KG, which it extends, can be viewed as a sequence of interrelated inference and reasoning procedures over a KG subset. The key challenges of those procedures are the discovery of methods that can simultaneously perform logical, comparative, quantitative and verification reasoning, and infer the answers that may not be explicitly present in a KG.

There are also practical differences arising from the benchmarks proposed for these tasks. First, unlike short artificially constructed questions with a single entity mention typical of the datasets for CQA-KG, such as CSQA [46] or ConvQuestions [8], the benchmark we propose for CER-KG makes no strict assumptions about the structure of the queries (as follows from Figure 1, the manually written queries in CER-KG can be arbitrarily long and include multiple entity mentions) or the nature of the resulting entities (unlike

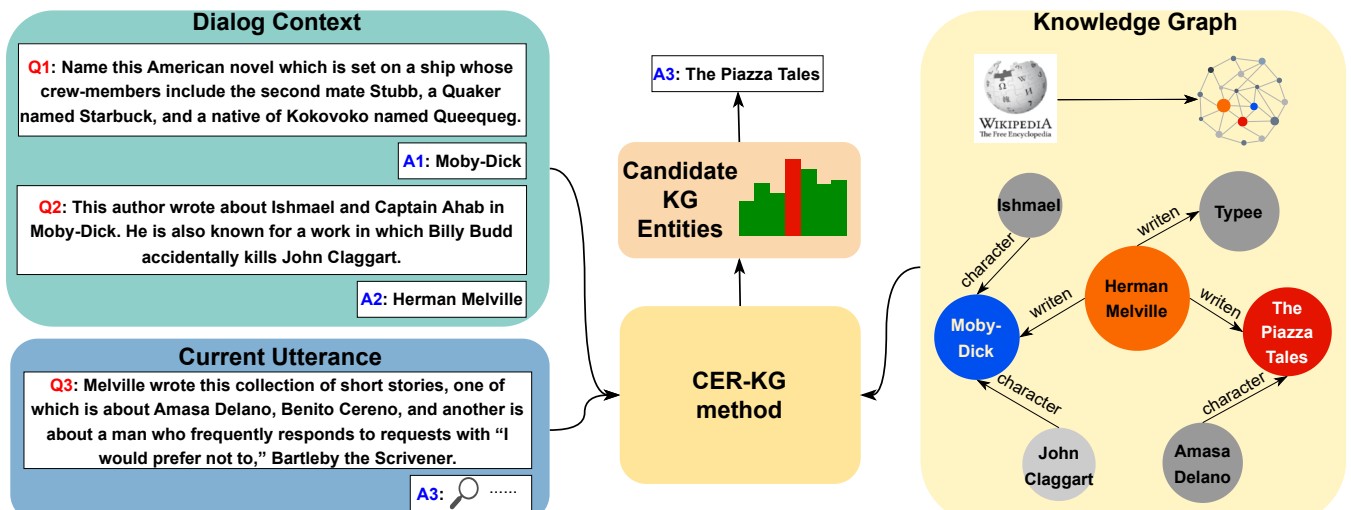

**Figure 1: Overview of the proposed task of Conversational Entity Retrieval from a KG (CER-KG).**

entity answers to questions in CSQA, which are restricted only to the object position of KG triples, resulting entity(ies) in CER-KG can be in the subject or object position of KG triplet(s)). Questions in CSQA, on the other hand, can have other answer types besides KG entities (e.g. numbers, true/false).

Overall, CER-KG complements CQA-KG in the landscape of methods that need to be developed for different types of conversational information-seeking interactions to enable its practical use.

As the first approach to CER-KG, we propose a **N**eural **A**rchitecture for **C**onversational **E**ntity **R**etrieval (NACER), a feature-based neural architecture to rank the candidate KG entities for each CER-KG dialog turn. Rather than taking distributed representations of the current dialog turn, dialog context, and a candidate KG entity to compute relevance signals internally, NACER directly utilizes diverse relevance signals as input features that capture the semantic and lexical similarities between a current dialog turn, preceding answer(s) and candidate entity's neighboring KG components, such as entities, categories, and literals. The candidate KG entities are then ranked according to their relevance scores computed by NACER. *Since NACER makes no restrictive assumptions about the dialog context and can be easily adapted to be used along with CQA-KG methods to generate responses at different turns of the same real-life information seeking dialog.*

To evaluate NACER and enable future research on CER-KG, we adapted QBLink [16], an existing benchmark for conversational reading comprehension of Wikipedia, to construct QBLink-KG, a CER-KG benchmark for DBpedia [29].[1]

## 2 RELATED WORK

### 2.1 Non-conversational entity retrieval from a KG

Benchmarks for non-conversational entity retrieval from a KG, such as DBPedia-Entity v2 [20], aim at finding an entity, an attribute of

---

[1]QBLink-KG and the source code of NACER and the baselines are publicly available at http://anonymized

an entity, or a list of entities in response to a keyword query or a question. Traditional IR methods proposed for this task [7, 39, 58] construct structured documents for each KG entity and aim to correctly weigh and aggregate lexical matches of the key query concepts in different fields of structured entity documents towards overall entity ranking score. The neural architectures proposed for this task range from feed-forward neural networks with attention [2] to transformers [6, 13, 17, 56] and aim to match dense representations of textual queries and KG entities.

### 2.2 QA and CQA over a KG

Prior research on QA over a KG independently studied simple and complex questions. Simple questions, such as those in the SimpleQuestions benchmark [3], correspond to a single KG triplet, in which the entity in the subject position is mentioned in a question and the entity in the object position is the answer. Existing approaches for simple QA over a KG can be grouped into two categories: end-to-end neural networks [21, 34] and pipelined approaches [33, 37, 40, 51, 57].

| Property | SQA | QA | CQA | ER | CER |
|---|---|---|---|---|---|
| Involves a multi-turn dialog | ✗ | ✗ | ✓ | ✗ | ✓ |
| Answer is present in a KG | ✓ | ✗ | ✗ | ✓ | ✓ |
| Answer is a KG entity | ✓ | ✗ | ✗ | ✓ | ✓ |
| Multiple types of answers or no answer | ✗ | ✓ | ✓ | ✗ | ✗ |
| Answer requires reasoning and/or inference | ✗ | ✓ | ✓ | ✗ | ✗ |
| Anaphora, co-references and ellipses | ✗ | ✗ | ✓ | ✗ | ✗ |

**Table 1: Summary of the key properties of Simple Question Answering (SQA), Complex Question Answering (QA), Conversational Question Answering (CQA), Entity Retrieval (ER) and Conversational Entity Retrieval (CER) over a KG.**

Complex QA over a KG has been well-studied in both non-conversational [5, 22, 32, 41, 47] and conversational [8, 18, 24, 25, 35, 48] settings. The major challenges of complex questions are that, besides entities, the answers to them can be yes/no, dates, numbers, or even no answer at all, and that answering them requires a multi-hop traversal of a KG, performing reasoning or comparison, aggregation, counting or set operations over a subset of a KG to discover the facts that may not be explicitly present in a KG. These challenges have been addressed with heuristic approaches [8], multi-hop inference [32, 47], reinforcement learning [25] and semantic parsing into an executable logical form [18, 22, 24, 35, 41, 48] or specialized language to represent the reasoning process [5].

The key properties of CER-KG, QA-KG and CQA-KG are summarized in Table 1, from which it follows that CER-KG methods cannot be evaluated on CQA-KG benchmarks, since the questions in them are not fully-formed due to the presence of anaphora, co-references and ellipsis. CQA-KG cannot be addressed using only IR methods due to their inability to perform advanced reasoning.

## 3 QBLINK-KG

QBLink-KG, our proposed benchmark for CER-KG, is adapted from QBLink [16], a high-quality, manually compiled benchmark for conversational reading comprehension over Wikipedia. QBLink consists of a short lead and a series of up to three queries, the answers to which are single named entities corresponding to the titles of Wikipedia articles. Formally, the task of CER-KG is to find out the correct answer (a KG entity) $a_k$ to a query $q_k$ in the $k$th dialog turn given the dialogue context, which includes all preceding queries $q_1, \ldots, q_{k-1}$ and their answers $a_1, \ldots, a_{k-1}$.

We used the English subset of the September 2021 DBpedia snapshot[2] as the target KG for QBLink-KG. Since DBpedia is constructed through information extraction from Wikipedia infoboxes [29], QBLink answers provided as the titles of Wikipedia articles can be easily converted into DBpedia entity URIs, if the corresponding entities exist in DBpedia.

| Filtering step | Train | Valid | Test |
|---|---|---|---|
| No filtering | 68,454 | 5,451 | 9,597 |
| wiki_page ≠ ∅ | 59,796 | 4,772 | 8,436 |
| Target entity ∈ $\mathcal{Y}$ | 14,586 | 1,100 | 1,682 |

**Table 2: Total number of queries in each split of the original QBLink and after each filtering step.**

Due to practical considerations, such as the limit on the model capacity imposed by the benchmark size, we only use the answer to the previous turn $a_{k-1}$ and query in the current turn $q_k$ in both the baselines and NACER. Nevertheless, the set of features used by NACER in Eq. 1 can in principle be expanded with the features that are based on $a_1, \ldots, a_{k-2}$ and $q_1, \ldots, q_{k-1}$.

QBLink cannot be utilized for CER-KG in its original form since knowledge graphs (even those derived from Wikipedia) contain significantly less information than Wikipedia. Specifically, a named entity that is an answer to a QBLink question may not exist as an entity in a given knowledge graph. To adapt QBLink to CER over

[2]https://databus.dbpedia.org/dbpedia/collections/dbpedia-snapshot-2021-09

| Statistic | Train | Valid | Test |
|---|---|---|---|
| Total words | 388,900 | 30,397 | 53,025 |
| Distinct words | 37,722 | 8,261 | 11,897 |
| Avg. words per query | 26.66 | 27.36 | 26.25 |

**Table 3: Statistics of QBLink-KG.**

DBpedia we performed two filtering steps illustrated in Table 2. First, we filtered out all QBLink queries that are unusable for the benchmark regardless of entity linking and candidate selection methods (i.e. all queries with an empty wiki_page field or those queries for which the answer does not correspond to a Wikipedia page or DBpedia entity). For the evaluation of NACER and the baselines with specific entity linking and candidate selection methods used in this work, we then filtered out the queries with the answers that do not belong to the set of candidate entities $\mathcal{Y}$ obtained with these methods.[3] The final statistics of QBLink-KG are shown in Table 3.

### 3.1 Entity linking and selection of candidate entities

Both NACER and the baselines utilize the same set of candidate entities $\mathcal{Y}$ generated based on the entities $e_l^1, \ldots, e_l^r$ linked from $q_k$, as shown in Figure 3. The entities linked to $q_k$ were obtained using the method proposed in [34][4], which proved to be effective for non-conversational simple QA over a KG. A set of candidate answer entities $\mathcal{Y}$ was obtained by including all other entities in the same triplets with the entities linked from $q_k$. To prevent an explosion of the set of candidate entities, we do not consider linked entities in $q_k$ with a degree greater than 100.

## 4 NACER

In order to identify the most effective types of relevance signals for CER-KG, we proposed NACER, a transparent, feature-based neural architecture for KG entity ranking. As shown in Figure 2, NACER has a modular architecture consisting of three major components: the encoding layer, the matching feature aggregation layers and the entity score computation layer.

### 4.1 Encoding Layer

**Features**. NACER computes the score of each candidate KG entity $y_i \in \mathcal{Y}$ based on the feature vector $\bar{y}_i$ constructed based on $q_k$, $a_{k-1}$ and $\mathcal{T}_i$, a set of all KG triplets that include $y_i$, as detailed in Table 4. The feature vector $\bar{y}_i$ for $y_i$ consists of the features derived using either semantic similarity function $f_e(\mathbf{a}, \mathbf{b})$ or lexical similarity function $f_w(a, b)$ based on: (1) lexical and distributed representations of KG structural components (entities, predicates, literals and categories) in $\mathcal{T}_i$; (2) lexical and distributed representations of $q_k$; (3) lexical and distributed representations of $a_{k-1}$:

$$\bar{y}_i = [\mathsf{ent}_e, \mathsf{pred}_e, \mathsf{lit}_e, \mathsf{cat}_e, \mathsf{ans}_e, \\ \mathsf{ent}_w, \mathsf{pred}_w, \mathsf{lit}_w, \mathsf{cat}_w, \mathsf{ans}_w]. \quad (1)$$

The first five features are calculated using $f_e$, while the last five features are calculated using $f_w$, as detailed in Table 4.

[3]to enable experiments with other entity linking and candidate entity selection methods, we will release both filtered and unfiltered versions of QBLink-KG.
[4]with the only difference is that the linked entities can be in the subject or object position

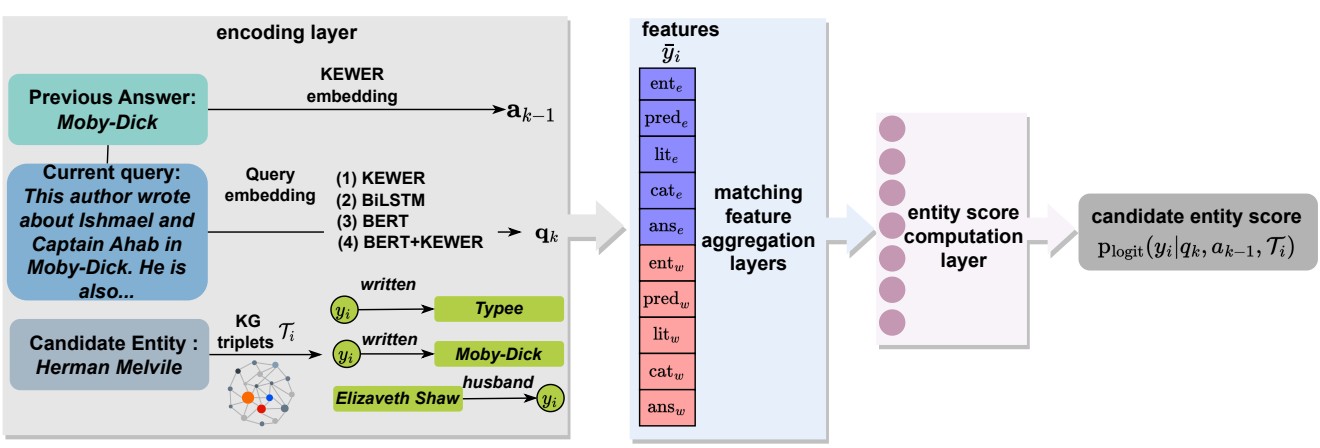

**Figure 2: Neural Architecture for Conversational Entity Retrieval from a Knowledge Graph.**

We experiment with three parametric and non-parametric variants of $f_e(\mathbf{a}, \mathbf{b})$ to determine the degree of similarity between the distributed representations of $\mathbf{a}$ and $\mathbf{b}$: (1) dot product $f_{e\text{-dot}}(\mathbf{a}, \mathbf{b}) = \mathbf{a}^\top \mathbf{b}$; (2) multiplicative interaction function $f_{e\text{-mult}}(\mathbf{a}, \mathbf{b}) = \mathbf{a}^\top \mathbf{W} \mathbf{b}$ with trainable parameter matrix $\mathbf{W}$; (3) additive interaction function $f_{e\text{-add}}(\mathbf{a}, \mathbf{b}) = \mathbf{v}^\top \tanh(\mathbf{W}_a \mathbf{a} + \mathbf{W}_b \mathbf{b})$ with trainable parameter vector $\mathbf{v}$ and matrices $\mathbf{W}_a$ and $\mathbf{W}_b$.

In addition, parameters $\mathbf{W}$ for the multiplicative interaction function, and $\mathbf{v}$, $\mathbf{W}_a$, $\mathbf{W}_b$ for the additive interaction function can be either shared between $\text{ent}_e, \text{pred}_e, \text{lit}_e, \text{cat}_e, \text{ans}_e$ features or trained for each feature individually.

$f_w(a, b)$ utilizes the bag-of-words representation of $a = \{a_1, \ldots, a_n\}$ and $b = \{b_1, \ldots, b_m\}$ to quantify lexical similarity as a sum of *smooth inverse frequencies* [1] of their overlapping terms:

$$f_w(a, b) = \sum_{w \in a \cap b} \frac{\lambda}{\lambda + n(w)}, \qquad (2)$$

where $\lambda$ is a hyper-parameter and $n(w)$ is the frequency of term $w$ in a KG.

**Embeddings**. We used the publicly available embeddings of words and KG structural components (entities, predicates, categories, and literals)[5] obtained using the KEWER method [38] in the encoding layer of NACER and for feature computation.

**Turn encoding methods**. The encoding layer first creates $\mathbf{a}_{k-1}$, a distributed representation of the preceding answer in the dialog, using KEWER. After that, it creates $\mathbf{q}_k$, a distributed representation of the $k$th turn in a CER-KG information-seeking dialog. We consider four options for dialog turn encoding: (1) **KEWER**: calculating the weighted mean of KEWER embeddings of the words and entities in $q_k$; (2) **BiLSTM**: embedding $q_k$ using a pre-trained BiLSTM with max-pooling [10]; (3) **BERT**: embedding $q_k$ with a pre-trained BERT [15]; (4) **BERT+KEWER**: embedding $q_k$ with the K-Adapter [54], a framework that allows integrating KG embeddings into a pre-trained BERT. Specifically, our K-Adapter injects the KG-specific information encoded in the KEWER embeddings into the representations created with pre-trained BERT.

―――――――
[5]https://academictorrents.com/details/4778f904ca10f059eaaf27bdd61f7f7fc93abc6e

## 4.2 Feature aggregation and score computation layers

Each candidate answer entity $y_i$ for the $k$th turn is then ranked based on its logit score:

$$p_{\text{logit}}(y_i | q_k, a_{k-1}, \mathcal{T}_i) =$$
$$\mathbf{w}_s^\top \sigma(\mathbf{W}_{a_2}^\top \sigma(\mathbf{W}_{a_1}^\top \bar{y}_i + \mathbf{b}_{a_1}) + \mathbf{b}_{a_2}) + b_s, \quad (3)$$

where $\mathbf{W}_{\{a_1, a_2\}}$ and $\mathbf{b}_{\{a_1, a_2\}}$ are the weights and biases in the matching feature aggregation layers (we use two in Eq. 3, but the number can vary); $\mathbf{w}_s$ is a weight vector of the size determined by the number of neurons in the final matching feature aggregation layer; $b_s$ is a scalar bias of the entity score computation layer, and $p_{\text{logit}}$ denotes a non-normalized logit probability, which is passed through the softmax function during the calculation of the loss.

## 4.3 Loss

Cross-entropy between one-hot distribution for the target entity $y_t$ and the entity logit score from Eq. (3) was used as the loss function.

## 5 EXPERIMENTAL SETUP

### 5.1 Baselines

**GENRE**. As the first baseline, we adapt GENRE [14], a method that fine-tunes BART [30] to retrieve entities by generating their surface forms token-by-token in an auto-regressive manner, to CER-KG. GENRE was shown to be superior to the entity retrieval methods using maximum-inner-product search over dense representations of queries and entities. In our adaptation, we consider the entire dialog context as a query, generate surface forms of answer entities and map them to entity URIs.

**KV-MemNN**. Memory networks (MemNNs) [55] are a class of differentiable models, which can perform simple inference over structured and unstructured knowledge. Key-value MemNNs [36], in which the memories are indexed by the keys, were shown to be effective at retrieving answers in text-based QA [36], non-conversational simple QA over a KG [3] and conversational QA over a KG [46]. As the baselines, we use the following two adaptations of the Key-Value

| Feature | Feature value | Feature description |
|---|---|---|
| $\mathtt{ent}_e$ | $f_e\left(\mathbf{q}_k, \frac{\sum_{(y_i,p_o,e_o)\in\mathcal{T}_i}\mathbf{e}_o + \sum_{(e_s,p_s,y_i)\in\mathcal{T}_i}\mathbf{e}_s}{|(y_i,p_o,e_o)\in\mathcal{T}_i| + |(e_s,p_s,y_i)\in\mathcal{T}_i|}\right)$ | semantic similarity between $\mathbf{q}_k$ and the mean of KEWER embeddings of KG entities that are either subject ($\mathbf{e}_s$) or object ($\mathbf{e}_o$) in the same triplet as $y_i$ |
| $\mathtt{pred}_e$ | $f_e\left(\mathbf{q}_k, \frac{\sum_{(s_j,p_j,o_j)\in\mathcal{T}_i}\mathbf{p}_j}{|(s_j,p_j,o_j)\in\mathcal{T}_i|}\right)$ | semantic similarity between $\mathbf{q}_k$ and the mean of KEWER embeddings of predicates $\mathbf{p}_j$ from the triplets in $\mathcal{T}_i$ |
| $\mathtt{lit}_e$ | $f_e\left(\mathbf{q}_k, \frac{\sum_{(y_i,p_j,l_j)\in\mathcal{T}_i}\mathbf{l}_j}{|(y_i,p_j,l_j)\in\mathcal{T}_i|}\right)$ | semantic similarity between $\mathbf{q}_k$ and the mean of embeddings $\mathbf{l}_j$ of literals from $\mathcal{T}_i$. $\mathbf{l}_j$ is calculated as the mean of KEWER embeddings of tokens in $l_j$ |
| $\mathtt{cat}_e$ | $f_e\left(\mathbf{q}_k, \frac{\sum_{(y_i,c_j)\in\mathcal{T}_i}\mathbf{c}_j}{|(y_i,c_j)\in\mathcal{T}_i|}\right)$ | semantic similarity between $\mathbf{q}_k$ and the mean of KEWER embeddings of categories $\mathbf{c}_j$ that $y_i$ belongs to |
| $\mathtt{ans}_e$ | $f_e\left(\mathbf{a}_{k-1}, \frac{\sum_{(y_i,p_j,o_j)\in\mathcal{T}_i}\mathbf{o}_j + \sum_{(e_s,p_s,y_i)\in\mathcal{T}_i}\mathbf{e}_s}{|(y_i,p_j,o_j)\in\mathcal{T}_i| + |(e_s,p_s,y_i)\in\mathcal{T}_i|}\right)$ | semantic similarity between $\mathbf{a}_{k-1}$ and the mean of KEWER embeddings of objects ($\mathbf{o}_j$) or subjects ($\mathbf{e}_s$) in the same triplets as $y_i$ ($o_j$ can be an entity, literal, or category) |
| $\mathtt{ent}_w$ | $\frac{\sum_{(y_i,p_o,e_o)\in\mathcal{T}_i}f_w(q_k,e_o) + \sum_{(e_s,p_s,y_i)\in\mathcal{T}_i}f_w(q_k,e_s)}{|(y_i,p_o,e_o)\in\mathcal{T}_i| + |(e_s,p_s,y_i)\in\mathcal{T}_i|}$ | average lexical similarity between $q_k$ and labels of KG entities that are either a subject ($e_s$) or an object ($e_o$) in the same triplet with $y_i$ |
| $\mathtt{pred}_w$ | $\frac{\sum_{(s_j,p_j,o_j)\in\mathcal{T}_i}f_w(q_k,p_j)}{|(s_j,p_j,o_j)\in\mathcal{T}_i|}$ | average lexical similarity between $q_k$ and labels of predicates $p_j$ from the triplets in $\mathcal{T}_i$ |
| $\mathtt{lit}_w$ | $\frac{\sum_{(y_i,p_j,l_j)\in\mathcal{T}_i}f_w(q_k,l_j)}{|(y_i,p_j,l_j)\in\mathcal{T}_i|}$ | average lexical similarity between $q_k$ and literals $l_j$ from $\mathcal{T}_i$ |
| $\mathtt{cat}_w$ | $\frac{\sum_{(y_i,c_j)\in\mathcal{T}_i}f_w(q_k,c_j)}{|(y_i,c_j)\in\mathcal{T}_i|}$ | average lexical similarity between $q_k$ and labels of all categories $c_j$ that $y_i$ belongs to |
| $\mathtt{ans}_w$ | $\frac{\sum_{(y_i,p_j,o_j)\in\mathcal{T}_i}f_w(a_{k-1},o_j) + \sum_{(e_s,p_s,y_i)\in\mathcal{T}_i}f_w(a_{k-1},e_s)}{|(y_i,p_j,o_j)\in\mathcal{T}_i| + |(e_s,p_s,y_i)\in\mathcal{T}_i|}$ | average lexical similarity between $a_{k-1}$ and objects ($o_j$) or subjects ($e_s$) in the same triplets as $y_i$ ($o_j$ can be entity, literal, or category) |

**Table 4: Semantic and lexical similarity features utilized by NACER for scoring candidate answer entities.**

Memory Network (KV-MemNN) [36] to CER-KG. These adaptations differ in the approaches used to fill $M$ key-value memory slots $(k_1, v_1), \ldots, (k_M, v_M)$.

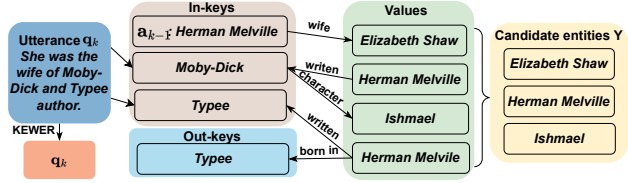

**Figure 3: Extraction of key-value memory slot pairs and candidate entities for the KV-MemNN baselines.**

The first approach (named KV-MemNN$_{\text{in}}$) uses the previous answer $a_{k-1}$ and $e_l^1, \ldots, e_l^r$, the entities linked from $q_k$, as keys $k_1, \ldots, k_M$ and entities in the same KG triplets as values $v_1, \ldots, v_M$. In this way, each key-value memory slot pair $(k_i, v_i)$ can be constructed from a single KG triplet, in which the subject or object $k_i$ is from the *in-key* set $\{a_{k-1}, e_l^1, \ldots, e_l^r\}$ and its opposing object or subject is used as a value $v_i$. Key-value memories are represented using the KEWER entity embeddings as $(\mathbf{k}_1, \mathbf{v}_1), \ldots, (\mathbf{k}_M, \mathbf{v}_M)$. The set of entities used as values $\{v_1, \ldots, v_M\}$ is considered as the candidate entities $y_1, \ldots, y_C$. Each candidate entity $y_i$ is scored using $\mathbf{q}_{H+1}$, the distributed representation of $q$ after $H$ hops over key-value memories and $\mathbf{y}_i$, the KEWER embedding of $y_i$, as $p_{\text{logit}}(y_i) = \mathbf{q}_{H+1}^{\top}\mathbf{y}_i$.

The second approach (named KV-MemNN$_{\text{out}}$) is identical in all aspects to KV-MemNN$_{\text{in}}$, except that the set of key-value memory slots $(k_1, v_1), \ldots, (k_M, v_M)$ is supplemented by the pairs $(k_i, v_i)$, where the value $v_i$ belongs to the set of candidate entities $\mathcal{Y} = \{y_1, \ldots, y_C\}$ as before, but the *out-key* $k_i$ is not necessarily from the set $\{a_{k-1}, e_l^1, \ldots, e_l^r\}$ and can be any neighbor of the candidate entity

$y_i$ (i.e. either a subject or an object in the triplet $\tau$ that contains $y_i$ as an object or a subject). Thus, the filling of memory slots is augmented in the following way. First, we consider an undirected knowledge graph $G$, where each subject-predicate-object triplet $(s, p, o)$ corresponds to the graph's $G$ undirected edge between the subject $s$ and object $o$. Second, an additional hop in $G$ is performed starting from the previously obtained value entities $v_i$ to obtain the *out-keys*.

Figure 3 illustrates the KV-MemNN$_{\text{in}}$ and KV-MemNN$_{\text{out}}$ approaches to filling the memory slots. Note that the set of candidate entities $\mathcal{Y}$ in both KV-MemNN$_{\text{in}}$ and KV-MemNN$_{\text{out}}$ is identical to the set of candidate entities used for our proposed NACER method, which allows for a fair comparison of the accuracy of NACER with KV-MemNN$_{\{\text{in,out}\}}$.

## 5.2 Hyperparameter settings and model design choices

Various hyperparameters are set to the values that have been demonstrated as effective in the existing literature [4, 28]. In Eq. (3), ReLU is used as a non-linearity function $\sigma$, and the numbers of neurons in the first and second matching feature aggregation layers of NACER are set to 20 and 10, respectively. The dimensionality of $\mathbf{v}$ in the additive interaction function is set to 512. We consider $n$-grams up to size 3 and set the number of candidate entities to 400 following [34]. Following [38], the term weighting parameter $\lambda$ in Eq. 2 is set to $3 \times 10^{-4}$. As the implementation of BiLSTM encoder with max pooling, we used V1 configuration of InferSent[6] encoder. We use the pre-trained bert-base-uncased model from the Hugging Face [7] as our BERT model. We fine-tune GENRE for 10 epochs using the training split of QBLink-KG and set the beam size to 10. We

---

[6]https://github.com/facebookresearch/InferSent
[7]https://huggingface.co/

| Method | $q_k$ encoding | $f_e$(a, b) | par. sharing | Hits@1 | R@1 | Hits@10 | R@10 | MRR |
|---|---|---|---|---|---|---|---|---|
| GENRE | - | - | - | 582 | 0.3460 | 856 | 0.5089 | 0.4002 |
| KV-MemNN$_{in}$ | KEWER | - | - | 991* | 0.5892* | 1496* | 0.8894* | 0.6905* |
| KV-MemNN$_{in}$ | BiLSTM | - | - | 854 | 0.5077 | 1449 | 0.8615 | 0.6269 |
| KV-MemNN$_{in}$ | BERT | - | - | 779 | 0.4631 | 1148 | 0.6825 | 0.5613 |
| KV-MemNN$_{in}$ | BERT+KEWER | - | - | 811 | 0.4822 | 1154 | 0.6861 | 0.6125 |
| KV-MemNN$_{out}$ | KEWER | - | - | 983 | 0.5844 | 1431 | 0.8507 | 0.6758 |
| KV-MemNN$_{out}$ | BiLSTM | - | - | 847 | 0.5035 | 1389 | 0.8258 | 0.6007 |
| KV-MemNN$_{out}$ | BERT | - | - | 765 | 0.4548 | 1131 | 0.6724 | 0.5512 |
| KV-MemNN$_{out}$ | BERT+KEWER | - | - | 802 | 0.4768 | 1143 | 0.6795 | 0.5587 |
| NACER | KEWER | dot | - | 648 | 0.3853 | 1314 | 0.7812 | 0.5172 |
| NACER | KEWER | mult | Y | 782 | 0.4649 | 1399 | 0.8317 | 0.5824 |
| NACER | KEWER | mult | N | 1016*‡ | 0.6040*‡ | 1567*‡ | 0.9316*‡ | 0.7164*‡ |
| NACER | KEWER | add | Y | 865 | 0.5143 | 1480 | 0.8799 | 0.6361 |
| NACER | KEWER | add | N | 977 | 0.5809 | 1533‡ | 0.9114‡ | 0.6967‡ |
| NACER | BiLSTM | mult | Y | 931 | 0.5535 | 1531‡ | 0.9102‡ | 0.6765 |
| NACER | BiLSTM | mult | N | 979 | 0.5820 | 1555‡ | 0.9245‡ | 0.7029‡ |
| NACER | BiLSTM | add | Y | 919 | 0.5464 | 1497‡ | 0.8900‡ | 0.6613 |
| NACER | BiLSTM | add | N | 1053*‡ | 0.6260*‡ | 1592*‡ | 0.9465*‡ | 0.7389*‡ |
| NACER | BERT | mult | Y | 807 | 0.4798 | 1439 | 0.8555 | 0.6067 |
| NACER | BERT | mult | N | 1016‡ | 0.6064‡ | 1573‡ | 0.9352‡ | 0.7178‡ |
| NACER | BERT | add | Y | 938 | 0.5577 | 1522‡ | 0.9049‡ | 0.6758 |
| NACER | BERT | add | N | 1095*‡ | 0.6510*‡ | 1600*‡ | 0.9512*‡ | **0.7658**\*‡ |
| NACER | BERT+KEWER | mult | Y | 979 | 0.5820 | 1553‡ | 0.9233‡ | 0.6993‡ |
| NACER | BERT+KEWER | mult | N | 1030‡ | 0.6124‡ | 1559‡ | 0.9269‡ | 0.7239‡ |
| NACER | BERT+KEWER | add | Y | 1048‡ | 0.6231‡ | 1569‡ | 0.9328‡ | 0.7297‡ |
| NACER | BERT+KEWER | add | N | **1121**\*‡ | **0.6665**\*‡ | **1602**\*‡ | **0.9524**\*‡ | 0.7575*‡ |

**Table 5: Accuracy of GENRE, and different variants of NACER and KV-MemNN on the test set of QBLink-KG. The largest value for each metric is highlighted in boldface. Each variant's best performance is indicated by $^*$. Statistical significance of the difference with KV-MemNN$_{in}$ and KEWER for $q_k$ encoding using the two-tailed paired Student's $t$-test with $p = 0.05$ is indicated by $^\ddagger$.**

compare the performance of KV-MemNN$_{in}$ and KV-MemNN$_{out}$ baselines using $H = 1, 2, 3, 4$ hops on the validation set and find out that both methods demonstrate the best performance when $H = 3$, which is the setting we used to report their results.

## 5.3 Training procedure

All variants of NACER and KV-MemNN were trained on the training split of QBLink-KG. To address overfitting, we utilized early stopping and save the model parameters resulting in the smallest loss on the validation set. Adam optimizer [27] with the learning rate of $10^{-3}$ was used to train all models, except NACER with $f_{e\text{-dot}}$, which was trained with the learning rate $10^{-5}$. KV-MemNN models were trained for 1000 epochs, and NACER models were trained for a maximum of 100 epochs, except NACER with $f_{e\text{-dot}}$ (1500 epochs) and NACER with $f_{e\text{-add}}$ and the KEWER embeddings-based turn encoder (500 epochs), since we found out that these configurations require a larger number of epochs to converge.

## 6 RESULTS

### 6.1 Retrieval accuracy

To examine different aspects of CER-KG and identify the types of methods that can be employed by effective solutions to it, we experimented with multiple dialog context encoders in combination with the key-value memory networks and NACER. We compare our proposed method with GENRE adapted to CER-KG. Results of different variants of NACER and KV-MemNN-based baselines along with GENRE on the test set of QBLink-KG are included in Table 5. Several conclusions can be drawn from these results.

First, the retrieval accuracy of NACER and KV-MemNN-based baselines surpasses GENRE adapted to CER-KG. While GENRE demonstrates proficiency in non-conversational entity retrieval, when straightforwardly extended to CER-KG, it falls short of the expectations, likely due to its failure to properly account for conversational context.

Second, NACER also consistently outperforms KV-MemNN-based baselines across all metrics in combination with any turn encoder type. The margin of the difference between the best configurations of NACER and KV-MemNN$_{in}$ ranges from 7% to 13 % for different metrics. Among all compared models, the NACER with the turn encoder using BERT and the KEWER-based K-Adapter, additive interaction function and no parameter sharing demonstrates the highest accuracy. We believe there are two major reasons behind this result. First, as a pre-trained language model, BERT already possesses rich knowledge acquired in an unsupervised manner from Wikipedia. This knowledge allows it to perform slightly better than BiLSTM as a turn encoder when most interaction functions are used to calculate the features capturing semantic similarity between distributed representations of the current turn and components of the KG surrounding the candidate entities. Second, our K-Adapter efficiently injects the KG-specific information captured by KEWER into BERT allowing it to better capture KG structure in the distributed representation of the current turn and the resulting features measuring its semantic similarity with the candidate entities, which in 17 out of 20 different configurations translates into additional

improvements in the range 0.1-21% over pre-trained BERT across different metrics. Finally, the superior performance of NACER over GENRE and KV-MemNN-based baselines can also be attributed to the need to take into account both semantic and lexical relevance signals, possibly due to the length of many queries in QBLink-KG.

Third, the dot product interaction function consistently resulted in the lowest accuracy among different semantic similarity functions utilized by NACER to compute the matching features. On the other hand, parametric multiplicative and additive interaction functions increase the capacity of NACER, which positively translates into its accuracy. Furthermore, parameter sharing of multiplicative and additive interaction functions has a consistently negative effect on the accuracy across all metrics. NACER paired with different types of turn encoders generally demonstrates better performance without parameter sharing.

Lastly, since KV-MemNN$_{out}$ consistently underperforms KV-Mem-NN$_{in}$ across all metrics, KV-MemNN does not benefit from the inclusion of the neighbors of candidate entities into its memory.

Overall, the above results indicate that the relevance signals pointing to the correct answer entity are mainly localized within a small neighborhood around that entity in a KG. As a result, finding the correct answer entity does not require the multi-hop inference procedure of the key-value memory networks. Instead, effective methods for CER-KG should focus on localizing, amplifying or attenuating with the right importance weights and combining diverse lexical and semantic matching signals in the answer entity's KG neighborhood.

## 6.2 Feature ablation

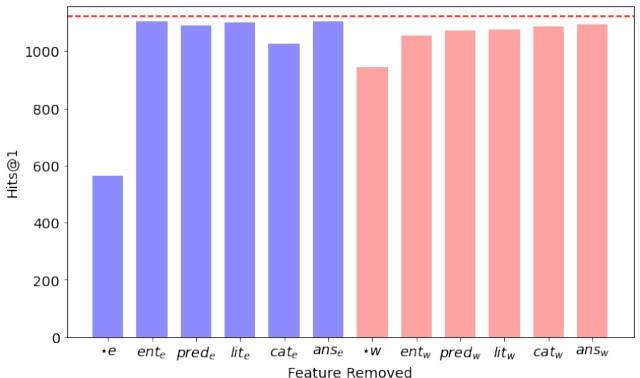

**Figure 4: Hits@1 of the best NACER configuration, when individual, all semantic and all lexical similarity features are removed. The red dotted line corresponds to Hits@1 when all features are used.**

To assess the relative importance of NACER features on its performance, we performed a feature ablation study. In this study, we removed one feature at a time by zero-masking the corresponding entry in $\bar{y}_i$ and retrained the best performing configuration of NACER (that uses BERT with KEWER-based K-Adapter as the turn encoder, additive interaction, and no parameter sharing). We also experimented with two additional configurations, in which all semantic similarity features ($*_e$) and all lexical similarity features ($*_w$)

were removed. The resulting Hits@1 values are shown in Figure 4. As follows from Figure 4, the performance drops significantly when either all semantic or all similarity features are removed, which indicates that both feature types are essential contributors to NACER's performance, with the semantic similarity features playing more important role than the lexical ones. Removal of most individual features (with a notable exception of $cat_e$ and $ent_w$) has a relatively smaller impact on Hits@1 of NACER. These results indicate that NACER effectively aggregates lexical and semantic matching features of candidate entities into their entity score.

## 6.3 Succes and failure analysis

The top 3 entities ranked by NACER and key-value memory network-based baselines in combination with different dialog context encoders are shown in Table 6. Examination of the results in this table also reveals qualitatively superior accuracy of NACER over the MemNet-based ranker. Specifically, regardless of the dialog context encoder, NACER was able to rank the correct entity as the top result for 2 out of 3 queries in the example information seeking dialog. Memory network-based ranker, on the other hand, was able to rank the correct entity in the top position only for 1 query and only with 1 dialog context encoder. Regardless of the dialog context encoder, NACER preserved the typical coherence of the top-ranked entities. Specifically, all entities top-ranked by NACER regardless of the context encoder for the first query in the dialog (Angela Carter, Sabine Huynh, Janez Menart and Peter Russell) are poets. All entities top ranked by both NACER in combination with BERT for the second query (The Waves, Orlando: A Biography and Mrs. Dalloway) and by the MemNet-based ranker in combination with BERT+KEWER adapter (Mrs. Dalloway, The Waves and Jacob's Room) are Virginia Wolf's novels, however, NACER was more precise at top ranking the correct answer entity. Similar observations can be made about the entities top-ranked by NACER and the MemNet-based ranker in combination with BERT. Jane Eyre, Villette, The Professor and Shirley are all Bronte's novels, however only NACER was able to correctly rank Jane Ayre as the top answer. Surprisingly, but consistent with the results in Table 5, using a weighted mean of KEWER embeddings as the dialog context encoder produces the most accurate results for the MemNet-based ranker. The top results for this configuration are typically consistent, unlike the combination of the MemNet-based ranker with BiLSTM encoder, but the MemNet-based ranker lacks precision. Overall ineffectiveness of the dialog context encoder based on the aggregation of KEWER embeddings can be attributed to the fact KEWER embeddings capture topical rather than typical similarity (e.g. Vanessa Bell is a sister of Virginia Woolf and Wise Children is a novel by Angela Carter).

## 7 CONCLUSION

In this paper, we introduced a novel task of CER-KG; QBLink-KG, the first benchmark for this task; and NACER, a feature-based neural architecture for CER-KG. Experimental results of NACER in combination with different types of dialog context encoder on the proposed benchmark indicate that localization and aggregation of lexical and semantic matching signals from the neighborhood of candidate answer entities in a KG is a more effective strategy

| Method | Turn | Top-3 answers and position of the correct answer | | | |
| --- | --- | --- | --- | --- | --- |
| | | **KEWER** | **BiLSTM** | **BERT** | **BERT+ KEWER** |
| **NACER** | Q1: Name this English author of novels like "The Passion of New Eve" and "Nights at the Circus", known especially for feminist reinterpretations of other works | 1. **Angela Carter** 2. Sabine Huynh 3. Janez Menart | 1. **Angela Carter** 2. Sabine Huynh 3. Janez Menart | 1. **Angela Carter** 2. Sabine Huynh 3. Janez Menart | 1. **Angela Carter** 2. Sabine Huynh 3. Peter Russell |
| | | 1 | 1 | 1 | 1 |
| | Q2: Carter wrote a libretto based on this Virginia Woolf novel, whose protagonist has affairs with Queen Elizabeth I and the princess Sasha and is mentored by Nicholas Greene while writing a long poem called "The Oak Tree" | 1. Freshwater (play) 2. The Waves 3. Vanessa Bell | 1. The Waves 2. Nights at the Circus 3. Wise Children | 1. The Waves 2. **Orlando: A Biography** 3. Mrs. Dalloway | 1. The Waves 2. **Orlando: A Biography** 3. The Magic Toyshop |
| | | 8 | 4 | 2 | 2 |
| | Q3: At her death, Carter left incomplete a sequel to this Charlotte Bronte novel. Carter's sequel would've been about Adele Varens, the adopted daughter of Mr. Rochester and this novel's title character | 1. **Jane Eyre** 2. Villette (novel) 3. Wise Children | 1. **Jane Eyre** 2. Jane Eyre (character) 3. Edward Rochester | 1. **Jane Eyre** 2. Villette (novel) 3. The Professor (novel) | 1. **Jane Eyre** 2. Villette (novel) 3. The Professor (novel) |
| | | 1 | 1 | 1 | |
| **KV-MemNN$_{in}$** | Q1: Name this English author of novels like "The Passion of New Eve" and "Nights at the Circus", known especially for feminist reinterpretations of other works | 1. Alamgir Hashmi 2. **Angela Carter** 3. Peter Russell | 1. Illusion and Reality 2. Sabine Huynh 3. Janez Menart | 1. Post-feminism 2. Janez Menart 3. Peter Russell | 1. Magic realism 2. Sabine Huynh 3. Janez Menart |
| | | 1 | 6 | 9 | 9 |
| | Q2: Carter wrote a libretto based on this Virginia Woolf novel, whose protagonist has affairs with Queen Elizabeth I and the princess Sasha and is mentored by Nicholas Greene while writing a long poem called "The Oak Tree" | 1. Mrs. Dalloway 2. Night and Day (novel) 3. Jacob's Room | 1. Hamza 2. Alt code 3. The Passion of New Eve | 1. Mrs. Dalloway 2. Nights at the Circus 3. Between the Acts | 1. Mrs. Dalloway 2. The Waves 3. Jacob's Room |
| | | 5 | 10+ | 10+ | 5 |
| | Q3: At her death, Carter left incomplete a sequel to this Charlotte Bronte novel. Carter's sequel would've been about Adele Varens, the adopted daughter of Mr. Rochester and this novel's title character | 1. **Jane Eyre** 2. The Professor (novel) 3. Villette (novel) | 1. Alt code 2. The Passion of New Eve 3. Hamza | 1. Shirley (novel) 2. The Professor (novel) 3. Villette (novel) | 1. Shirley (novel) 2. The Professor (novel) 3. Villette (novel) |
| | | 1 | 10+ | 10+ | 10+ |

**Table 6: Top-3 entities returned by NACER and KV-MemNN$_{in}$ baselines in combination with KEWER, BiLSTM, BERT and BERT with KEWER $K$-Adapter context encoders along with the rank of the correct entity for queries in the same QBLink-KG information seeking dialog. The correct answer entity is highlighted in boldface, if present in the top 3 results.**

to address this task, than multi-hop inference and auto-regeressive answer generation.

In conclusion, we would like to outline possible avenues for future work. First, the performance of NACER and the key-value network-based baselines is equally significantly affected by the methods utilized for entity linking to the current query and candidate entity selection steps, even though these steps are external to NACER and the baselines. Alternative approaches to those used in this work for

these steps may improve or decrease the reported results and warrant further investigation in future work.

Similarly, the performance of NACER and the baselines may depend on several factors related to the target KG, such as its freshness and completeness. No aspects of NACER and the employed methods for entity linking and candidate entity selection are specific to DBpedia, however, adapting QBLink-KG to other knowledge graphs (e.g. Wikidata) and evaluating the performance NACER on this adaptation is another possible avenue for future work.

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
