# OpenReview forum: "Benchmark and Neural Architecture for Conversational Entity Retrieval from a Knowledge Graph"
_ACM.org/TheWebConf/2024/Conference — TheWebConf24_

### Official Review · Reviewer_Kb2n · 2023-11-24

**Novelty:** 5
**Technical Quality:** 5

**Review:**

The subject of the article is very interesting from many perspectives. The authors propose a new dataset and approach to evaluate Entity Retrieval in a conversational setting. Using an available dataset, i.e. QBLlink, they build a new benchmark to evaluate the capacity of systems to extract the appropriate entity that answers a given query in a multi-turn QA system.

It seems that the paper’s goal is clear, and the approaches to conduct the experiments are sound. Nonetheless, the presentation in the paper, especially from the perspective of a researcher more focused on Conversational Search systems, sounds a bit hard to follow. More specifically, in section 3, when describing the filtering phase of the QBLink original dataset, the authors filter based on a set of candidate entities Y. This part seems unclear and may require further specification.

Also, the part regarding the baselines, section 5.1, was very specific and, therefore, uneasy to understand for a more general audience.
Overall the topic of the paper is interesting. It is a good idea to release a new dataset for the evaluation of entity retrieval in a Conversational setting. This could be used also in different tasks such as query expansion and query rewriting.

The evaluation of the paper is good overall, with some revisions concerning the presentation of the experiments, it can achieve an even better quality.

**Questions:**

As stated on the review, one question concerns the third filtering step. I was not able to guess what you refer to with the “set of candidate entities y” in section 3.

Being more familiar with Conversational Search systems, it was a bit hard to follow the more technical parts of the paper. If you’d like to make it more accessible, maybe you can try to stress better the task and its different phases before describing in detail what you did.

Last, for future developments, you could try to test the Nacer for some different tasks, such as query expansion.

**Ethics Review Description:**

does not apply

**Reviewer Confidence:**

3: The reviewer is confident but not certain that the evaluation is correct

**Scope:**

4: The work is relevant to the Web and to the track, and is of broad interest to the community

---

### Official Review · Reviewer_NKqk · 2023-11-25

**Novelty:** 5
**Technical Quality:** 4

**Review:**

In this paper, the authors extend an entity-retrieval dataset QBLink-KG to a conversational setting (CER-KG).The task is to determine the correct answer (a Knowledge Graph entity) to a query in a given dialog turn, considering the context of all preceding queries and their answers. The authors used the English subset of the September 2021 DBpedia snapshot as the target Knowledge Graph for QBLink-KG.

The authors also propose a baseline (NACER)  which computes various features derived using neural embeddings to retrieve the entities in the conversational setting.

The authors miss important relevant dataset: Wizard of Wikipedia [1] which focuses on conversational models that leverage knowledge from Wikipedia. It primarily involves training conversational agents to effectively use and reference Wikipedia knowledge during conversations. This approach emphasizes the integration of vast unstructured textual data into conversational AI. While this dataset is not directly designed to retrieve entities, it can be adapted easily to link the DBPedia entity to Wikipedia page the answer is coming from.

[1] Dinan, Emily, Stephen Roller, Kurt Shuster, Angela Fan, Michael Auli, and Jason Weston. "Wizard of wikipedia: Knowledge-powered conversational agents." arXiv preprint arXiv:1811.01241 (2018).

Strengths:
1. The proposed approach structured to consider the KG aspects, rather than just textual content.
2. The features computed are more human interpretable.
3. Paper is well written and easy to read.

Limitations:
1. The authors miss relevant works.
2. The dataset has a narrow scope which works only DBPedia and entity-oriented conversations.
3. Comparison to larger generative models is missing.

**Questions:**

1. What are the novel aspects of your dataset and baseline methods compared to Wizard of Wikipedia and their baselines?
2. How do the few-shot and in-context learning using larger LLMs compare to the proposed feature-based appraoches?
3. It is unclear how the QBLink dataset was extended to the conversational setting. How were the conversations created?

**Reviewer Confidence:**

4: The reviewer is certain that the evaluation is correct and very familiar with the relevant literature

**Scope:**

4: The work is relevant to the Web and to the track, and is of broad interest to the community

---

### Official Review · Reviewer_YShU · 2023-11-27

**Novelty:** 5
**Technical Quality:** 4

**Review:**

The paper introduces a new task, Conversational Entity Retrieval from a Knowledge Graph (CER-KG), and proposes NACER, a model built on handcrafted features designed for this task.

The paper employs a complex notation that hampers readability. Understanding how NACER performs retrieval is challenging due to the abundance of subscripts, varied fonts, and symbols used.

The authors suggest that while NACER, in its current form, could theoretically incorporate features based on previous answers and queries, practical constraints limit this to utilizing only the prior turn's answer (ak−1) and the current turn's query (qk). As a result, the task seems less conversational and more aligned with classical KG Entity Retrieval (ER).

Do W{a1,a2} and b{a1,a2} refer to the first and second answers? If not, the notation remains unclear.

The experiments primarily compare NACER against two baseline models. Table 5 seems more focused on analyzing the components of NACER rather than benchmarking against the current state-of-the-art (SOTA). Additionally, the most robust comparison, KV-MemNN, is eight years old.

Table 6 is anecdotal and not really informative. While intriguing, it might be better suited as an appendix rather than occupying an entire page in the main body of the paper

**Questions:**

Do W{a1,a2} and b{a1,a2} refer to the first and second answers? If not, the notation remains unclear.

Are the baselines chosen indeed the most effective approaches, considering particularly that NACER is based only on the last utterance and can be easily mapped on classical ER?

**Ethics Review Description:**

No ethical issues

**Reviewer Confidence:**

3: The reviewer is confident but not certain that the evaluation is correct

**Scope:**

4: The work is relevant to the Web and to the track, and is of broad interest to the community

---

### Official Review · Reviewer_MZxL · 2023-11-29

**Novelty:** 5
**Technical Quality:** 5

**Review:**

Summary:
The paper introduces Conversational Entity Retrieval from a Knowledge Graph (CER-KG), a novel information retrieval task where user queries in a conversational setting depend on previous dialog turns that involve Knowledge Graph (KG) entities. The authors propose a Neural Architecture for Conversational Entity Retrieval (NACER), which ranks KG entities based on their relevance to the dialog context. NACER uses a feature-based approach to consider lexical and semantic matching signals between dialog turn, preceding answers, and KG entities. For evaluation, the authors adapted an existing benchmark, QBLink, to create QBLink-KG, a CER-KG benchmark for DBpedia.

Strengths:
1. The paper addresses a gap in conversational information retrieval by focusing on entity retrieval from KGs in a dialog setting, which is a unique and relevant area given the advancements in conversational AI.
2.  NACER's design is robust, considering a wide range of lexical and semantic features from dialog contexts and KG entities, indicating a thorough approach to the problem.
3.  Adapting the QBLink benchmark to create QBLink-KG for DBpedia is a practical approach, facilitating further research in this new area.

Weaknesses:
1.  The paper primarily introduces the architecture and the benchmark, but lacks the source code of the benchmark.
2. The intricate design of NACER, while comprehensive, might pose challenges in implementation and optimization, especially in the dynamic KG.

**Questions:**

1. How does NACER handle ambiguities and evolving contexts in prolonged conversational settings?
2. Are there plans to extend the evaluation of NACER beyond the QBLink-KG benchmark, possibly in more diverse real-world datasets?
3. How does the performance of NACER compare with other established IR systems, especially in handling complex KG queries?

**Reviewer Confidence:**

2: The reviewer is willing to defend the evaluation, but it is likely that the reviewer did not understand parts of the paper

**Scope:**

4: The work is relevant to the Web and to the track, and is of broad interest to the community

---

### Official Review · Reviewer_wroa · 2023-11-30

**Novelty:** 3
**Technical Quality:** 3

**Review:**

In this paper, a novel task named Conversational Entity Retrieval from a Knowledge Graph (CER-KG) is introduced. This task involves treating conversational contexts and previous answers as queries, with the goal of retrieving entities in the Knowledge Graph (KG). The authors also construct a benchmarking dataset based on QB-Link and develop feature-based Learning to Rank (LTR) models for reranking candidates identified through entity-linking. Despite these contributions, I find that the drawbacks of this work outweigh its strengths.

**Pros:**

- The paper is well-written, providing clear descriptions of technical details, including figures and tables.
- The proposed technical contribution demonstrates the best performance, as evaluated by the adopted metrics.
- The authors conduct a thorough analysis of the results, encompassing both successes and failures.
- Introducing a new task could potentially interest other researchers in the field.

**Cons:**

- While the proposed task emphasizes entity "retrieval," the NACER approach is essentially focused on "reranking" candidates identified by an entity linker. This mismatch may mislead readers about the nature of this work.
- The technical contribution primarily revolves around identifying the best LTR feature based on different text encoders with limited novelty.
- The connection between CER-KG and CQA-KG, and their complementary effects, as discussed in the introduction, seems far-fetched. The utility of CER-KG for CQA-KG is not clearly justified.
- The authors assert that "NACER makes no restrictive assumptions about the dialog context" in section 1 but contradict this by stating, "Due to practical considerations, such as the limit on the model capacity imposed by the benchmark size, we only use the answer to the previous turn 𝑎_{𝑘−1} and query in the current turn 𝑞_𝑘 in both the baselines and NACER."
- The use of the baseline GENRE appears inappropriate. Candidates are exposed to the model during the ranking stage for KV-MemNN and NACER but not for GENRE. Additionally, it seems that GENRE is not optimized on the training set of the collection, placing it and other models (LM-based generative ones) at a significant disadvantage. While generative models theoretically handle longer and variable-length conversations, the experimental settings overlook these factors.

**Questions:**

- Could the authors provide further discussion on the connection between CER-KG and CQA-KG and elaborate on its importance? Additionally, why is the task CER-KG important independently? In what scenarios would a user pose a series of questions where the answers involve multiple connected entities?

- What specific "practical considerations" led the authors to use only the answer to the previous turn 𝑎_{𝑘−1} and the query in the current turn 𝑞_𝑘 in both the baselines and NACER? In the case of GENRE, considering the entire context as the input query is possible. Is it more about NACER's limitations in handling a variable number of turns in a conversation? If only the last turn is considered, it significantly diminishes the utility of the entire conversational setting.

**Reviewer Confidence:**

4: The reviewer is certain that the evaluation is correct and very familiar with the relevant literature

**Scope:**

4: The work is relevant to the Web and to the track, and is of broad interest to the community

---

### Decision · Program_Chairs · 2024-01-22

**Decision:**

Accept

**Comment:**

This paper introduces a new task, Conversational Entity Retrieval from a Knowledge Graph, and proposes a model leveraging handcrafted features designed for this task.

 The paper was reviewed by five reviewers. The paper has clearly some merits. Most reviewers agree on the technical quality and novely of the papers, but they also raise some comments still requirint a proper explanation. Please clarify this point in the camera-ready copy.